# Prevalence of Adverse Childhood Experiences in Students with Emotional and Behavioral Disorders in Special Education Schools from a Multi-Informant Perspective

**DOI:** 10.3390/ijerph19063411

**Published:** 2022-03-14

**Authors:** Evelyne C. P. Offerman, Michiel W. Asselman, Floor Bolling, Petra Helmond, Geert-Jan J. M. Stams, Ramón J. L. Lindauer

**Affiliations:** 1Orion, Special Education, Bijlmerdreef 1289-2, 1103 TV Amsterdam, The Netherlands; m.asselman@orion.nl (M.W.A.); f.bolling@orion.nl (F.B.); 2Levvel, Academic Center for Child and Adolescent Psychiatry, Meibergdreef 5, 1105 AZ Amsterdam, The Netherlands; p.helmond@levvel.nl (P.H.); r.lindauer@levvel.nl (R.J.L.L.); 3Department of Child Development and Education, University of Amsterdam, Nieuwe Achtergracht 127, 1018 WS Amsterdam, The Netherlands; g.j.j.m.stams@uva.nl; 4Department of Child and Adolescent Psychiatry, Amsterdam University Medical Centre, University of Amsterdam, Meibergdreef 5, 1105 AZ Amsterdam, The Netherlands

**Keywords:** adverse childhood experiences, prevalence, multi-informant, emotional and behavioral disorders, students, special education schools, trauma-informed education

## Abstract

Adverse childhood experiences (ACEs) are associated with an increased risk of developing severe emotional and behavioral problems; however, little research is published on ACEs for students with emotional and behavioral disorders (EBD) in special education (SE) schools. We therefore systematically explored the prevalence, type and timing of ACEs in these students from five urban SE schools in the Netherlands (*M_age_* = 11.58 years; 85.1% boys) from a multi-informant perspective, using students’ self-reports (*n* = 169), parent reports (*n* = 95) and school files (*n* = 172). Almost all students experienced at least one ACE (96.4% self-reports, 89.5% parent reports, 95.4% school files), and more than half experienced four or more ACEs (74.5% self-reports, 62.7% parent reports, 59.9% school files). A large majority of students experienced maltreatment, which often co-occurred with household challenges and community stressors. Additionally, 45.9% of the students experienced their first ACE before the age of 4. Students with EBD in SE who live in poverty or in single-parent households were more likely to report multiple ACEs. Knowledge of the prevalence of ACEs may help understand the severe problems and poor long-term outcomes of students with EBD in SE.

## 1. Introduction

Adverse childhood experiences (ACEs), such as abuse and parental mental illness, are associated with an increased risk of developing emotional and behavioral problems [1,2,3]. Moreover, ACEs are associated with many of the problems affecting students with emotional behavioral disorders (EBD) in special education (SE) schools, but very little research is published on this topic [4]. In the Netherlands, students with EBD account for 37% of all students in separate SE schools and represent a subgroup of students with EBD with the most intensive needs. These students experience severe and persistent problems in interpersonal relationships, self-regulation, social competence and academic development and put high demands on schools in serving their emotional, behavioral and educational needs [5,6,7]. Despite access to SE schools and ongoing mental health services [8,9], the long-term academic, relational and health outcomes of this SE population are poor and lead to high costs (financial and otherwise) for individuals and society [7,10,11]. Knowledge of the prevalence of ACEs may be crucial for understanding the complex problems of students with EBD in SE schools and may help SE schools be more responsive to their needs. Therefore, as a first step to ultimately improve the long-term perspectives of these students, we explore the prevalence of ACEs in students with EBD in separate SE schools.

### 1.1. Students with EBD in Special Education

According to the International Convention on the Rights of the Child, all children have the right to an education that enables their personalities, talents and mental and physical abilities to develop to their fullest potential [12]. To fulfill this key role in society and ensure such a perspective for all students, SE support is available in schools for students with additional educational needs due to cognitive, health, physical, emotional and/or behavioral problems. Internationally, SE support is organized in a continuum, increasing in restrictiveness [13,14], striving to educate all students in regular education classrooms with typically developing peers of the same age as much as possible [15,16,17]. In the Netherlands, SE is organized in a ‘stepped care principle’, offering support from light to intensive. National criteria for admission to a separate SE school have not been in force since 2014. Students can be referred if the additional SE support services in regular schools are proven insufficient. Professionals from a regional authority then decide on eligibility for SE placement.

In Western nations such as the USA and the UK, students with EBD comprise one of the largest groups that are educated along the continuum of SE support, totaling 0.5% to 2% of the general population of students aged 4–21 years [10,18]. EBD is internationally used to describe students who have elevated levels of emotional, behavioral or social difficulties compared to their peers and who are therefore in need of SE support [11,18]. These students experience severe and persistent difficulties in building or maintaining satisfactory interpersonal relationships with peers and teachers. They have difficulty with the self-regulation of emotions and behavior. They also display externalizing behaviors such as aggression and, often less visible, internalizing problems, such as a negative self-image as well as low well-being [8,9,10,13,18,19,20]. EBD also refers to emotional and behavioral problems that are classified as various psychiatric disorders such as attention deficit hyperactivity disorder (ADHD), autism spectrum disorder (ASD) and oppositional defiant disorder (ODD). Overall, students with EBD have poor long-term perspectives [13]. Examples of the specific long-term challenges for students with EBD include school drop-out, reduced school performance, poor social and relational outcomes, a higher risk of juvenile offending, disconnection from the community and high levels of unemployment [8,9,10,20,21].

In the Netherlands, students with EBD can be characterized in line with the aforementioned international descriptions. Two-thirds of all students with EBD are ultimately placed in separate SE schools [5]. For students with EBD who have the most intensive needs [8,22], these schools offer education in relatively small classes of 10–15 students [23] and provide additional support from paraprofessionals, school psychologists and youth healthcare [5,9]. An often unintended effect of the previously mentioned stepped-care principle is that an accumulation of negative experiences at their previous school (usually in regular education), such as rejection of peers, conflicts with teachers, failure in adaptation and cognitive development, often precedes placement in a SE school for EBD [22,24]. Consequently, although placement in SE is intended to be temporary, SE schools in the Netherlands are often a ‘last resort’ [22]. Due to the severity of their problems, the majority of students with EBD continue their educational careers in SE schools after initial placement [25]. SE schools for EBD are assigned a great responsibility in preparing their students in becoming full members of society. However, the complex needs of these students on an emotional, behavioral and educational level [5,6,26,27] and a severe shortage of qualified teachers [7,10,28,29,30,31] make it challenging for SE schools to fulfill this role. So far, little progress is being made in designing interventions for students with EBD with the most complex needs and etiologies [7]. Therefore, new approaches are needed to understand and serve their complex needs and research to guide policy and practice [7,10,32].

### 1.2. Adverse Childhood Experiences

Recent studies showed that ACEs are an important factor for understanding well-being and lifelong health outcomes [33,34]. ACEs are broadly defined as single or chronic exposures in the environment during childhood (0–18 years) that are distressing, potentially harmful and traumatic [35,36,37,38]. Prior research showed that ACEs are associated with a greater risk of negative outcomes such as poor school performance, school drop-out, juvenile involvement with the criminal justice system and job-related problems [2,3,39,40,41,42]. In 25 years of ACEs research, one consistent finding is that ACEs are very common among school-aged youth. The systematic review of Carlson and colleagues [43] on the prevalence of ACEs demonstrated that almost two-thirds of school-aged youth worldwide experienced at least one ACE. Moreover, ACEs tend to co-occur, and they have a dose–response relationship with their associated problems [2,3,39,40,41,42]. The accumulation of ACEs in high doses during critical and sensitive periods early in life, without buffering factors such as nurturing caregivers and safe and stable environments, could lead to toxic stress as the stress response systems in the brain and body are activated excessively or prolonged [3]. Therefore, experiencing multiple ACEs exponentially increases the chances of poor outcomes [3,36]. ACEs are associated with many of the emotional, behavioral and academic problems that students with EBD in SE encounter. These include aggression and anxiety, psychiatric problems according to the DSM, such as ADHD, ASD and post-traumatic stress disorder (PTSD), as well as significant delays in cognitive development and academic achievement [1,2,38,39,41,42,44,45,46,47,48,49,50,51].

### 1.3. Adverse Childhood Experience in Students with EBD in SE

Knowledge of the prevalence of ACEs in the lives of students with EBD in SE schools is a crucial first step in better understanding the potential role of ACEs and the emotional, behavioral and school problems that these students encounter. An accumulation of ACEs can be expected for students with EBD in SE schools since vulnerable populations are particularly more at risk of experiencing ACEs [35,43]. Moreover, students with EBD who are placed in separate SE schools often grow up in vulnerable families that are characterized by poverty, single parenthood and a non-western cultural background [9,10,20,21,52]. While some of these demographic characteristics could be considered an ACE in itself (e.g., poverty), they are also known to be risk factors for experiencing ACEs, both in the general population [52,53,54] and in children and youth with EBD [13].

Despite the expected associations between ACEs and emotional and behavioral problems, in a previous systematic review, we showed that ACEs have hardly been assessed for students with EBD in SE schools [4]. Based on the few studies available, an indication was found for an association between ACEs, primarily maltreatment, and placement in SE in general [55,56,57,58]. Furthermore, of the few ACEs assessed, prevalence rates of 31–86% for abuse and neglect were found in students with EBD in SE [59,60].

Despite tentative indications on elevated prevalence rates for ACEs for students with EBD in SE schools, these indications are based on only a few studies, and various uncertainties remain. First, previous studies included a small range of ACEs, assessing primarily maltreatment, although the original ACE framework is broader than maltreatment only. In addition, the concept of ACEs evolved over recent decades from the original ten ACEs measured by Felitti and colleagues [36] towards an extended range of experiences within and outside the family context [37,61]. ACEs from this ‘broadened framework’ were shown to have the same or similar outcomes as the 10 original ACEs [61,62,63]. Moreover, adding ACEs concerning peer and community stressors improved the prediction of psychological distress in adults and youth [40,63]. A second source of uncertainty is that most studies on ACEs in students with EBD in SE depended on a single informant perspective [4], which could lead to an underestimation of prevalence rates. There are several reasons for this underestimation. Depending entirely on self-reports may lead to underreporting of ACEs [64], and parents may be unaware of all the adverse events their children have experienced [65,66]. Additionally, self-reported and parent-reported measures are inherently subjective and prone to recall bias [3]. Although professional reports could provide a more objective assessment, underreporting in these informants also remains common [3,67]. Therefore, to determine how students with EBD in SE schools are affected by ACEs, a broad ACEs framework and multi-informed perspective are needed.

### 1.4. The Present Study

Exploring the prevalence of ACEs is a first and important step in understanding the potential key role of ACEs in the severe and complex problems of students with EBD in SE. From a trauma-informed approach, understanding the effect of ACEs, trauma and toxic stress could help SE schools change the lens through which students’ emotional and behavioral difficulties are perceived and be more responsive to their complex needs. In this explorative study with a sample of 174 students with EBD in SE schools, we addressed the following questions: (1) What is the prevalence, type and timing of ACEs in students with EBD in SE schools from a multi-informant perspective? and (2) Does the prevalence of ACEs differ significantly between subgroups based on demographic characteristics (sex, age, parents’ country of birth, family size, household composition, educational level of the parents, economic status) and DSM diagnoses?

## 2. Materials and Methods

### 2.1. Sample

Our sample consisted of 174 students with EBD aged 8–18 years (*M_age_* = 11.58 years, 85.1% boys) in five urban primary and secondary separate SE schools. These schools operate under the auspices of a special education foundation in the Netherlands. The majority of students in the sample started their school careers in regular education schools (84.5%). Based on previous intelligence research reports in the students’ school files, we established that the IQ score of our sample was roughly estimated as normal to borderline intellectual functioning (*M* = 89.58, *SD* = 17.85). However, there was a large variance in IQ scores of our sample (range 49 to 137). The school files demonstrated that in the assessment of the IQ, the WISC-III NL was predominantly used (74.5%), followed by the WPPSI-III-NL (11.8%).

Most students were given a psychological evaluation and met the criteria for one or more diagnoses according to the diagnostic and statistical manual of mental disorders (DSM-IV or DSM-5). More than half of the students used medication related to their DSM diagnoses. The majority of the students received some form of mental health support, either individually or family-oriented. The biological parents of the participating students represented more than 10 ethnic backgrounds, with Dutch, Surinamese, Moroccan, Antillean, Ghanaian and Turkish being the most common. Table 1 and Table 2 provide a full overview of the demographic- and student characteristics. All descriptives are based on school file reports.

### 2.2. Design and Procedures

The current study is part of an ongoing research project investigating factors that possibly influence the onset and maintenance of behavioral and emotional problems and mental illness of students in Dutch urban SE schools for EBD. We used a descriptive retrospective cross-sectional design and a multi-informant perspective.

Data was collected between the years 2017 and 2020. The study was approved by the Ethics Review Board of the University of Amsterdam (2017-CDE-7603). We briefed primary caregivers, teachers and students about the project and asked for active and documented informed consent. The consent rate of parents for child participation was 35%. The flowchart (Figure 1) shows the number of respondents per informant and the reasons for non-participation. The final number of participants was *n* = 169 for students and *n* = 95 for parents and had access to *n* = 172 school files.

To ensure the most optimal number of participants and their degree of disclosure, we adapted the methods of ACE assessment to the specific needs of our sample. The questions had little jargon, and each student had an individual assessment supported by the presence of a professional with sufficient time to complete the questionnaire. Parents scored the questionnaire either online or on paper at home. In addition, based on a codebook developed by the authors, we scored the participating students’ school files. We used all available reports from previous and current schools settings and care settings, such as school progress reports, daily school journals, diagnostic reports, psychiatric reports and youth health care service reports. Research assistants were trained in how to use the codebook.

### 2.3. Measures

#### 2.3.1. Measures of Student Self-Reports

ACEs in the student self-reports were measured by the Dutch version of the Life Events Checklist (LEC). The LEC is part of the Clinician-Administered PTSD Scale for Children and Adolescents interview (CAPS-CA) from 8 to 16 years old [68,69,70]. We used the LEC because it includes a broad spectrum of potentially traumatic life events, and since the manner of questioning was expected not to provoke feelings of shame, fear and loyalty conflict towards parents in the students. The LEC represents 25 ACEs or life events (Appendix A). Answer categories were ‘It happened to me personally’, ‘I have witnessed it happen to someone else’; ‘I have learned about it happening to someone close to me’; ‘I am not sure if it applies to me’; and/or ‘It does not apply to me’. The number of ACEs was determined by a total score of the events that students reported being directly exposed to (it happened to me personally and/or I have witnessed it happen to someone else = 1). These two types of exposure can be considered the most severe type of exposure [71]. Additionally, students completed four questions derived from an ACE questionnaire [72] to measure three household challenges that were not represented in the LEC: parental substance abuse (alcohol and drugs), parental mental health problems and suicide or attempted suicide of a member of the household [33]. For these additional questions, students could report whether it was happening to them right now (score 1), not now, but in the past (score 1) or not at all (score 0). We determined the number of additional ACEs by counting the items scored as 1. To measure the internal consistency of the LEC combined with the additional ACE questions, we used Kuder–Richardson’s formula (KR20) due to the dichotomous answering categories (0/1). The measures of self-reports had a good level of internal consistency (KR20 = 0.81). We calculated the sum of ACEs of the student self-reports if at least 26 of the 28 items were completed. Six students had a maximum of two missing items; all other students completed the questionnaire.

#### 2.3.2. Measures of Parent Reports

ACEs in the parent reports were also measured by the LEC (Appendix A), with scores ranging between 0–25 ACEs [68,69,70]. Answer categories were ‘It happened to my child personally’, ‘My child has witnessed it happen to someone else’; ‘My child has learned about it happening to someone close to him/her’; ‘I am not sure if it applies to my child’; and/or ‘It does not apply to my child’. The internal consistency was acceptable (KR20 = 0.73). We calculated the sum of ACEs of the parent reports if at least 23 of the 25 items were completed. Therefore, 12 parent reports with one or two missing items were included, and four parent reports were excluded (see Figure 1). We directed all items of the LEC, for both students’ self-reports and parent reports, into categories based on the work of Asmundson and Afifi [35].

#### 2.3.3. Measures of School File Reports

The codebook used in the school file reports included 10 ACEs from the original Adverse Childhood Experiences Study [36], 12 ACEs from an expanded ACEs framework and 3 ACEs from the category ‘other’ [35,73,74,75,76,77,78,79,80,81,82,83,84,85,86,87,88,89,90,91]. Despite the use of an extensive ACEs framework, we encountered an even wider range of ACEs while examining the school files, such as a near-drowning experience or the death of a sibling, which we scored under the category ‘other’. One of the 12 ACEs from an expanded ACEs framework is ‘medical trauma/stressful medical event(s) of the child’. We added this ACE to the expanded framework because of its unexpectedly frequent occurrence (almost 20%) in our EBD sample in SE schools. Appendix B provide an overview and operationalization of the original and expanded ACEs in the school files.

Each item in the codebook was scored and substantiated with text from the school file. ACEs that were substantiated and ACEs that were reported in the school files according to the operationalization in the codebook (unsubstantiated ACEs) were scored as present (‘1′) [57,92]. If no information was found or the available information did not meet the criteria in the operationalization, the ACE was scored as absent (‘0′). Furthermore, we rated the quality of the school files using a low, moderate and high score (Appendix C). Most school files appeared to be of low (69.8%) or moderate (27.9%) quality. We reported the timing of the first ACE each student encountered in the categories 0–4 years old, 4–8 years, 8–12 years, 12–16 years, 16 years and older or unspecified. Each present ACE was checked for accuracy by a different research assistant than the coder while transporting to SPSS. For 10.5% of the school files, the inter-rater reliability was calculated for the complete codebook (162 items), including the ACEs. With 86.4%, the inter-rater reliability was considered as good.

### 2.4. Data Analysis

Statistical Package for the Social Sciences (SPSS; version 27) was used for all data analyses. Descriptive and inferential statistics were evaluated, including frequencies, means, standard deviations and bivariate associations. The data were not normally distributed; therefore, non-parametric tests were used in the present study (Spearman R, Mann–Whitney U, Kruskal–Wallis H and the Friedman test). Bivariate associations between the number of ACEs reported by the three different informants were tested using the Spearman R correlation. School files showed a small correlation (*r_s_* = 0.18, *p* = 0.020) with self-reports and a moderate correlation (*r_s_* = 0.33, *p* = 0.001) with parent reports. No significant correlation was found between students’ self-reports and parent reports), although the association was in the expected direction (*r_s_* = 0.15, *p* = 0.146).

Because of the difference in the number of parent reports (*n* = 95) compared to self-reports (*n* = 169) and school files (*n* = 172), we firstly explored the potential bias between parents who completed their questionnaires and those who did not. We compared their demographic characteristics (household composition, country of birth) and the present ACEs in the category household challenges (parental separation or divorce; parental mental health problems; economic hardship; bad accident or physical illness of a parent) using the Mann–Whitney U test. There were no significant differences between the two parent groups for these variables. Furthermore, we did not find a significant difference in the number of ACEs between the students of which ACE scores of all informants (self-report, parent report and school files) were available (*n* = 92) and the students whose parent’s reports were missing (students’ self-reports *n* = 77, school files *n* = 80). Therefore, we ran all analyses for our full sample.

## 3. Results

### 3.1. Prevalence, Type and Timing of ACEs

Nearly all students with EBD in SE schools had experienced at least one ACE as self-reported by the students (96.4%), in the parent reports (89.5%) and in the school files (95.3%). Moreover, the majority of the students experienced four or more ACEs based on self-reports (74.4%), parent reports (52.7%) and school files (59.9%). Furthermore, a substantial proportion of students with EBD in SE schools experienced eight or more ACEs according to their self-reports (40.2%), parent reports (12.7%) and school files (20.3%). An overview of the prevalence of ACEs and the mean number of ACEs by the informant is presented in Table 3.

We ran a Mann–Whitney U test to determine if there were differences in the mean rank of ACEs for each informant with regard to sex (male/female) and school type (primary or secondary SE school). No significant differences were found between the number of ACEs and sex for all informants (self-reports *U* = 1835, *z* = −0.11, *p* = 0.916; parent reports *U* = 577, *z* = −0.024, *p* = 0.809; school files *U* = 1575, *z* = −1.39, *p* = 0.165). Furthermore, no significant school type differences were found in the mean rank of ACEs for self-reports (*U* = 2684, *z* = −1.86, *p* = 0.063) and parent reports (*U* = 990, *z* = −0.02, *p* = 0.987). However, in the school files the mean rank of ACEs for primary school students (mean rank = 98.46) was significantly higher than for secondary school students (mean rank = 63.58, *U* = 1982, *z* = −4.39, *p* < 0.001).

A Friedman test was run to determine if there were differences between the quality of the school files, the school type (primary or secondary) and the total number of ACEs in school files. A significant difference was found, χ2(2) = 246.764, *p* < 0.0005. Pairwise comparisons were performed with a Bonferroni correction for multiple comparisons. These post hoc analyses showed significant differences in the ‘quality of the school files’ (Mdn = 0.00) in relation to ‘the number of ACEs in school files’ (Mdn = 4.00), *p* < 0.001, as well as in ‘the quality of school files’ in relation to ‘school type’ (Mdn = 1.00), *p* < 0.0005 and in ‘school type’ in relation to ‘the number of ACEs in school files’, *p* < 0.001. Therefore, the median difference in the number of ACEs for primary and secondary school files could be attributed to the quality of school files, which was significantly lower in secondary schools. This median difference could therefore be disregarded in the analysis.

Prevalence rates of the various types of ACEs are reported separately for each informant in Table 4 (self-reports and parent reports) and Table 5 (school files). According to all informants, a large majority of students experienced maltreatment: 84% for self-reports, 74.7% for parent reports, 69% for school files. Additionally, all informants frequently reported bullying from the category of peer victimization (60.4% for self-reports; 63.2 for parent reports; 32.1% for school files). ACEs that were not assessed by all informants but were reported by one or two of the informants for at least 20% of the students were economic hardship, parental separation or divorce, death of someone close, traffic accidents, other serious accidents, being stalked, physical assault with a weapon, fire/explosion and other (not specified) severe or frightening events.

Co-occurrence of different types of ACEs was found using Spearman’s R correlation, both within ACE categories, e.g., different types of maltreatment, and across ACE categories. Specifically for school files, where the most ACEs in the category household challenges were measured, we noted that maltreatment often co-occurred with parental mental health problems, parental physical illness, economic hardship and parental substance abuse. We found that many ACEs co-occurred with at least four other ACEs for all informants. Appendix D, Appendix E and Appendix F provide an overview of these bivariate associations.

In the school files, the timing of the first ACE of each student was registered, if available. We found that almost half of the students experienced their first ACE before the age of 4 (45.9%) (Figure 2).

To improve the comparison of the prevalence rates in our study to previous ACE studies, we determined the prevalence of the 10 original ACEs from Felitti and colleagues [36] in our sample, based on the school files reports. The parents’ and students’ reports did not include all these 10 ACEs; therefore, the multi-informant perspective could not be used here. The results concerning the 10 original ACEs (*M* = 2.41, *SD* = 1.95, range 0–7) showed that the majority of students (79.7%) experienced at least one ACE, and 24.4% experienced four or more.

### 3.2. Demographic Characteristics, Diagnoses and the Prevalence of ACEs

Table 6 provide an overview of five demographic characteristics (i.e., risk factors) in relation to the mean number of ACEs in the school files. Both the presence of living in a single-parent household (*U* = 1685, *z* = −2.21, *p* = 0.027) and economic hardship (*U* = 1073, *z* = −5.07 *p* < 0.001) showed a significantly higher mean rank of ACEs compared to students who grew up in families without this risk factor. The relationship between an accumulation of risk factors and the mean rank of ACEs in the school files was not statistically significant (*r*_s_ = 0.07). However, we found that students with ≥ 2 risk factors had a significantly higher mean rank of ACEs in the school files compared with students with 0–1 risk factors (*U* = 2539, *z* = −2.27 *p* = 0.023).

Additionally, we examined the differences in ACE prevalence in students for the various DSM diagnoses outlined in Table 1, taking comorbidity into account. A Kruskal–Wallis H test was used. We found only small differences, which were non-significant (single diagnosis: self report χ2(2) = 6.627, *p* = 0.469; parent report χ2(2) = 7.918, *p* = 0.340, school files χ2(2) = 17.650, *p* = 0.014; comorbid diagnoses: χ2(2) = 2.042, *p* = 0.360; parent report χ2(2) = 0.985, *p* = 0.611, school file χ2(2) = 3.190, *p* = 0.203).

## 4. Discussion

In this study, we explored the prevalence, types and timing of ACEs in students with EBD in separate SE schools. We assessed ACEs from a broad ACEs framework and a multi-informant perspective (student self-reports, parent reports, school files) to address the following research questions: (1) What is the prevalence, type and timing of ACEs in students with EBD in SE schools from a multi-informant perspective? and (2) Does the prevalence of ACEs differ significantly between subgroups based on demographic characteristics (sex, age, parents’ country of birth, family size, household composition, educational level of the parents, economic status) and DSM diagnoses? Our results showed that, from a multi-informant perspective, ACEs are very common in the lives of students with EBD in SE schools. Almost all students experienced at least one ACE (96.4% self-reports, 89.5% parent reports, 95.3% school files), and many students experienced a strong accumulation of ACEs at a very young age. More than half of the students experienced four ACEs or more (74.4% self-reports, 52.7% parent reports, 59.9% school files). Because of a possible selection bias in our sample, the fact that not all ACEs were assessed for all informants and the fact that the quality of school files was low-moderate, we expect our prevalence rates to be an underestimation.

Concerning the types of ACEs, the majority of students experienced some form of maltreatment, with physical abuse (71.0% self-reports, 57.9% parent reports, 34.5% school files) and emotional abuse (44.4% self-reports, 57.9% parent reports, 19.6% school files) most frequently reported. ACEs often co-occurred with at least four other ACEs, both within and across categories. With regard to the timing of ACEs, almost half of the students with EBD in Dutch urban SE schools experienced their first ACE before the age of four.

Although we expected our prevalence rates to be elevated compared to the general population, the number of ACEs in students with EBD in SE seems strikingly high. Comparing our findings to previous studies is difficult because our study is the first to systematically explore ACEs in these students with EBD in SE schools. Nevertheless, our results indicate that the number of students that experience at least one ACE in our sample (96.4% self-reports, 89.5% parent reports, 95.3% school files) is high compared to previous ACEs studies in school-aged youth (i.e., ≤18 years), which reported almost two-thirds of youth worldwide to experience at least one ACE (ranging from 41% to 97% in USA studies and 19% to 83% for studies outside of the USA) [43]. Moreover, in a Dutch sample, 45.3% of the students in regular education schools (10 to 11 years old) reported at least one of the 10 original ACEs [36], and 6.5% reported four or more ACEs [74]. In our sample, 79.7% of the students had at least one ACE reported in their school file, and 24.4% had four or more from the 10 original ACEs. These results indicate that our sample of students with EBD in SE has elevated ACE prevalence scores regardless of the number of ACEs assessed.

Our expectation that prevalence rates in our sample are much higher than in the general population is underlined by the fact that for many ACEs, specifically forms of maltreatment and household challenges, preliminary indications from previous studies [4] were confirmed, showing much higher prevalence rates in students with EBD in SE schools than in the general population [43,93]. For example, our study indicates a prevalence for physical abuse of 71% in self-reports, 57.9% in parent reports and 34.5% in school files versus 2.9% in a general population [94]. The high ACE prevalence rates we found in students’ school files based on the 10 original ACEs are comparable to previous studies involving other vulnerable populations, such as the overall population with EBD problems (including students in regular education), children with intellectual disabilities, youth at risk for residential placement and male and female juvenile offenders [95,96,97,98]. The prevalence rates reported in this study did not differ for various DSM-related disorders. The risk of experiencing an accumulation of ACEs seems to be particularly high for students with EBD in SE who live in a single-parent household or in economic poverty, a finding in line with previous studies in a general population as well as in the overall population with EBD problems (including students in regular education) [53,54,95]. Compared to the general population of the Netherlands, many of the students in our sample were raised in families with multiple demographic family risk factors, e.g., elevated numbers of single-parent households, economic hardship and larger family size [99,100,101]. Presumably, students with EBD in SE schools with multiple demographic family risk factors experience more ACEs, although more research is needed in larger samples to confirm this finding with more certainty.

### 4.1. Clinical Implications

The results of our study underline the need to increase awareness of ACEs in stakeholders such as the government, kindergarten, school boards, school referral systems and school professionals. Additionally, our findings of the timing of ACEs emphasize the need to work jointly towards interventions before children enter school (in the Netherlands at the age of four) to prevent ACEs or their further accumulation in order to reduce their potentially harmful effects. As a start, the assessment of ACEs should be included in each diagnostic evaluation when emotional and behavioral problems arise in students.

When ACEs are assessed for students with EBD in SE, a broad framework should be considered. Our findings indicate that assessors should at least pay attention to household challenges and parental stress and well-being, in addition to maltreatment, as these are known to be associated [102,103]. Furthermore, other ACEs outside the family, such as bullying, accidents and medical trauma, should be taken into account; these may not be immediately obvious or expected. Another important aspect in the assessment of ACEs is the use of a multi-informant perspective, as our study underlines that these different informants provide additional information on both the number, type and timing of ACEs [64,66,104]. For schools, it is particularly important to have high-quality files, as in our sample, the majority of school files were of low–moderate quality and potentially missed valuable information on students’ ACEs.

Furthermore, we recommend exploring trauma-informed education as a promising contribution for students with EBD in SE schools. Trauma-informed education could change the lens through which students’ emotional and behavioral difficulties or disorders are perceived by professionals and can improve students’ ability to learn, which may contribute to the reduction of physical aggression, referrals, and trauma symptoms [105]. Therefore, trauma-informed education is assumed to promote healing and resilience when students have experienced adversity [1,38,106]. Meanwhile, trauma-informed education could reduce stress and burnout in teachers and consequently preserve healthy teachers for a very vulnerable school population.

### 4.2. Limitations

This study has a number of limitations that should be mentioned. We expect these limitations to contribute to an underestimation of our prevalence rates. First of all, our sample was possibly not representative of the full population of participating schools. Although additional efforts were made to reach all parents for active, informed consent, more than half of the parents did not respond. For those parents who actively refused participation, one of the main reasons was that participation was thought to be too stressful because the parent was overburdened or because of the trauma experiences/many diagnostic evaluations (often trauma-related) of their child. Secondly, although the LEC is a user-friendly questionnaire that is widely used in trauma-informed clinical practice where there is considerable variability in the temporal stability of self-reported trauma exposure [107], the LEC is not yet validated for children and adolescents. Furthermore, the LEC does not cover the range of ACEs that are commonly assessed in ACEs research; it leaves out ACEs such as divorce, emotional neglect and economic hardship. Lastly, the overall quality of school files was often low or moderate at best. In many cases, a number of reports were missing, particularly in secondary schools. Overall, it is likely that the number of ACEs reported both on the LEC and in school files is an underestimation of all ACEs that were experienced by the students.

### 4.3. Future Research

Despite these limitations, our study could serve as a baseline in the Netherlands and internationally for research on ACEs in students with EBD in SE schools. As a next step, future research should focus on the assessment of ACEs in students with EBD receiving SE in larger and representative samples in the Netherlands and abroad. Just as in clinical work, it is recommended that a multi-informant perspective and a broad ACE framework be used. Furthermore, ACEs should be related to the specific problems students with EBD in SE have and to their short- and long term developmental outcomes. In this respect, not only the number of ACEs but also the type, timing, frequency and severity of ACEs should be taken into account, as the impact of ACEs can significantly differ because of these aspects [3,34,108]. Moreover, the potential mediating role of students’ resilience and other present risk or buffering biopsychosocial factors on outcomes should be included in future research of this specific population. Additionally, as ACEs within the family context are overwhelmingly present in our population, with high numbers of parental mental health problems, divorce, parental substance abuse, an intergenerational approach needs to be explored. From the current knowledge of the high prevalence of ACEs in students with EBD in SE schools, future research should also direct the attention to the siblings as well, as all children in a home could be at risk for the ACEs that are encountered by students with EBD.

## 5. Conclusions

Our study contributes to the literature in a number of ways. First, to the best of our knowledge, this study is the first to systematically explore the prevalence of ACEs in students with EBD in SE schools, a vulnerable, high-risk school population with poor short-term and long-term outcomes. Second, we used a broad framework that included ACEs associated with child maltreatment, household dysfunction, peer victimization and community stressors. Third, we used a multi-informant perspective, with students’ self-reports, parent reports and school files. Our results suggest that ACEs are highly prevalent in the lives of Dutch urban students with EBD in SE schools and start at a very young age. Awareness of the high prevalence of ACEs from a multi-informant perspective widens our perspective to look beyond current EBD symptoms and regard the students’ severe and persistent problems from a holistic and trauma-informed perspective. It urges us to address ACEs as early as possible to prevent accumulation and a long-lasting impact. Trauma-informed education could be a promising approach to tailor education to the needs of ACE-exposed students and to those who work with them and can therefore contribute to optimal future perspectives for students with EBD in SE schools.

## Figures and Tables

**Figure 1 ijerph-19-03411-f001:**
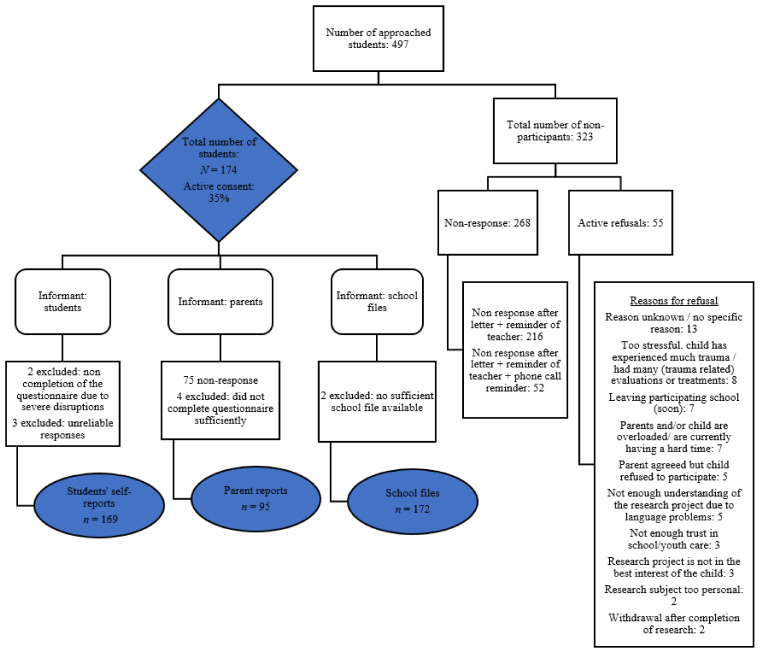
Flowchart of research participation.

**Figure 2 ijerph-19-03411-f002:**
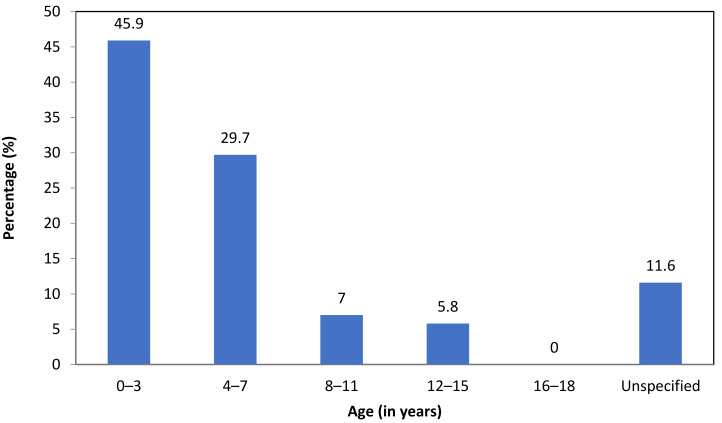
Timing of the first ACE in school files (*n* = 172) Note. For 11.6% of the students, the timing of the first ACE was not registered and therefore added to the category unspecified.

**Table 1 ijerph-19-03411-t001:** Demographic characteristics of the sample (*N* = 174).

Characteristics	*M* (*SD*) (Range)	%
School type		
Primary special education (age 4–12 years)		65.5
Secondary special education (age 12–18 years)		34.5
Sex		
Male		85.1
Female		14.9
Age		
Primary special education	10.08 (1.47)	
Secondary special education	14.43 (1.45)	
Primary and secondary special education	11.58 (2.54)	
IQ score	89.58 (17.85) (49–137)	
Family size ^a^	2.50 (1.23)	
Mother’s country of birth		
Netherlands		49.4
Western country		9.2
Non-western country		36.2
Missing		5.2
Father’s country of birth		
Netherlands		33.9
Western country		5.7
Non-western country		39.1
Missing		21.3
Household composition		
Original household		33.9
Blended household		8.0
Single parent household		43.7
Other		13.2
Missing		1.1
Educational level (of the primary income earner)		
Low		13.2
Moderate		27.6
High		14.9
Missing		43.7

^a^ One student was living in a group home at the time of this research and was therefore excluded.

**Table 2 ijerph-19-03411-t002:** Student characteristics: school switches, psychological evaluation and therapy (*N* = 174).

Characteristics	Variables	%
School switches ^a^	0	0
	1	32.2
	2	40.2
	3	19.0
	≥4	8.6
Type of school switch ᵇ	RE-RE	23.0
	RE-SE	84.5
	SE-SE	51.1
	SE-RE	6.3
Psychological (diagnostic) evaluation	Yes	98.9
	Missing	1.1
Diagnostic classification ^c^	Yes	84.5
	No	15.5
Diagnostic classification according to DSM-IV or 5	Autism spectrum disorders ^d^	35.1
	Attention-deficit/hyperactivity disorder	35.1
	Oppositional defiant disorder	16.7
	Disruptive behavior disorder NOS	9.2
	Specific learning disorder	6.9
	Post-traumatic stress disorder	6.3
	Reactive attachment disorder	5.7
	Borderline intellectual functioning	5.7
	Other	16.5
Comorbidity	No diagnosis	15.5
	1 diagnosis	43.1
	2 or more diagnoses	41.4
Medication use related to DSM diagnosis	Yes	53.5
	No	42.0
	Missing	4.6
Child therapy	Yes	87.4
	No	11.5
	Missing	1.1
Family-oriented social work and therapy	Yes	85.6
	No	13.2
	Missing	1.1

^a^*M* = 2.13 (*SD =* 1.24 Min/Max 1–10). Corrected for school switch from primary to secondary school (*n* = 60) ᵇ RE = Regular Education; SE = Special Education. ^c^ ‘No’ includes: The student has not (or not yet) completed a psychological (diagnostic) evaluation or is currently being diagnosed; the student has completed a psychological evaluation, but no diagnosis was found; the school does not have access to the student’s psychological evaluation, and therefore it is not known whether or not the student has a diagnosis. ^d^ Includes one student diagnosed with a multiple complex developmental disorder.

**Table 3 ijerph-19-03411-t003:** Number of ACEs per informant.

ACEs	Students’ Self-Reports (%) *n* = 169	Parent Reports (%) *n* = 95	School Files (%) *n* = 172
0	3.6	10.5	4.7
1	3.6	11.6	12.2
2	9.5	10.5	13.4
3	8.9	14.7	9.9
4–7	34.3	40.0	39.6
≥8	40.2	12.7	20.3
≥1	96.4	89.5	95.3
≥4	74.4	52.7	59.9
*M (SD)*	6.89 (4.40)	3.91 (2.84)	4.70 (3.27)
Min/Max	0–25	0–14	0–14
Maximum number of ACEs	28	25	25

**Table 4 ijerph-19-03411-t004:** Prevalence of ACEs in students’ self-reports (*n* = 169) and parent reports (*n* = 95).

Included ACEs	Self-Reports (%)	Parent Reports (%)
Maltreatment	84.0	74.7
Neglect	18.3	5.3
Supervisory neglect	14.2	3.2
Physical neglect	5.9	4.2
Abuse	81.1	74.7
Physical abuse	71.0	57.9
Emotional abuse	44.4	57.9
Physical assault with a weapon	27.2	8.4
Sexual abuse	10.7	2.1
Forced to be somewhere	8.3	0
Witnessed people having sex or porn	7.7	7.4
Domestic violence	24.3	29.5
Household challenges and/or peer victimization	85.2	76.8
Bullying	60.4	63.2
Death of someone close	58.6	37.9
Forced into doing something (non-sexual)	18.3	9.5
Police arrest of a family member	17.2	13.7
Parental mental health problems	12.4	
Parental substance abuse	10.1	
Suicide or attempted suicide of a household member	7.7	
Community stressors	33.1	8.4
Stalked ^a^	21.3	4.2
Experienced war or neighborhood violence	15.4	4.2
Experienced shooting ᵇ	11.2	1.1
Accident	75.7	46.3
Other serious accident	55.0	19.0
Traffic accident	47.9	24.2
Fire/explosion	27.2	10.5
Natural disaster	8.3	4.2
Exposure to hazardous substances	5.9	0
Other	62.7	30.5
Hurting someone severely	47.9	13.7
Other severe or frightening events	23.7	10.5
Witnessed other people injured or dead	17.2	7.4
Serious illness or near to dear through severe injury	11.8	8.4

^a,^ᵇ Context of the experience was not specified.

**Table 5 ijerph-19-03411-t005:** Prevalence of ACEs in school files (*n* = 172).

Included ACEs	Percentage (%)
Maltreatment	69.0
Neglect (total)	53.4
Physical/supervisory neglect	33.9
Psychological neglect	20.7
Medical neglect	13.3
Educational neglect	8.0
Unspecified	2.3
Abuse (total)	41.4
Physical abuse	34.5
Emotional abuse	19.6
Sexual abuse	5.1
Unspecified	1.7
Domestic violence	29.3
Household challenges	82.0
Parental separation or divorce	47.7
Parental mental health problems	33.3
Separation from parents	25.9
Economic hardship	20.1
Many (sudden) relocations	17.8
Serious accident or physical illness of a parent	17.2
Substance abuse	9.7
Parental absence	9.2
Parental death	4.6
Parental incarceration	4.0
Peer victimization	40.2
Bullying	32.1
Negative school experiences	12.1
Community stressors	3.5
Victim of neighborhood violence	3.5
Other	40.1
Other severe or frightening events ^a^	29.7
Medical trauma	18.4

^a^ Students experienced one or more (up to three) other ACEs.

**Table 6 ijerph-19-03411-t006:** Demographic characteristics in relation to the mean number of ACEs in school files.

Demographic Risk Factors	*n*	*M*	*M* Rank	Test	*U/H*	*z*	*p*
Household composition ^a^	133			MWU	1685	−2.21	0.027
Original household	57	3.12	58.55				
Single parent household	76	4.74	73.34				
Mother’s country of birth	163			MWU	3269	−0.17	0.869
Born in NL	84	4.61	82.59				
Not born in NL	79	4.32	81.37				
Father’s country of birth	135			MWU	1802	−1.93	0.054
Born in NL	77	3.86	60.56				
Not born in NL	58	4.53	73.60				
Parents’ country of birth	132			MWY	1788	−1.24	0.215
Born in NL	50	3.94	61.25				
≥1 not born in NL	82	4.40	69.70				
Educational level ᵇ	96			KWH	5.619		0.060
Low	23	4.65	49.39				
Moderate	48	5.35	53.75				
High	25	3.68	37.60				
Family size	171			MWU	3164	−1.26	0.208
0–2 children in household	99	4.82	90.04				
≥3 children in household	72	3.99	80.55				
Economic hardship ^c^	172			MWU	1073	−5.07	<0.001
Not present	137	3.91	76.83				
Present	35	6.80	124.34				
Accumulated risk factors ^d^	160			MWU	2539	−2.27	0.023
0–1	81	4.00	72.34				
≥2	79	4.81	88.87				

Note. MWU = Mann–Whitney U test. KWH = Kruskal–Wallis H test. NL = The Netherlands. ^a^ Compared without the ACE for parental separation or divorce. ᵇ Educational level of the primary income earner. ^c^ Compared without the ACE economic hardship. ^d^ Risk factors are: single-parent household, mother not born in NL, low education level, ≥3 children in the household and economic hardship present as ACE).

## Data Availability

The data presented in this study are available on request from the corresponding author. The data are not publicly available due to privacy.

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
