# Peer review of "Prevalence of Adverse Childhood Experiences in Students with Emotional and Behavioral Disorders in Special Education Schools from a Multi-Informant Perspective"

_ijerph, 2022, doi:10.3390/ijerph19063411_

Round 1

Reviewer 1 Report

The Authors take up an interesting topic of great cognitive and practical value, which lies within the journal's thematic scope. Understanding the scale of adverse childhood experiences (ACEs) in the group of children and adolescents with selected disorders is necessary for undertaking preventive measures, among others, targeting potential risk factors.

I still have a few more remarks listed below. I hope that the Authors will want to include them in their work, or consider them from the perspective of subsequent works devoted to the analyzed problem.

Introduction

This part is well-developed and constitutes a solid justification of the research problem. It is based on rich references. Notes on this part:

  1. Please specify which groups of people in the Dutch education system are covered by EBD. Are these only diagnostic cases that occurred in the Authors’ research, or are there also other ones? Are there any formal criteria for this group?
  2. The EBD category includes very different cases of disorders with a specific etiology and diagnostic criteria. Consider referring to studies showing the extent of ACEs occurrence in these different groups (eg. ASD).
  3. Please justify why the analyses focus on demographic characteristics but they do not include the diagnostic category type of disorder. It seems to me that it can significantly differentiate ACEs, as mentioned above.

Materials and methods

I do not see any significant shortcomings in this part. It is carefully prepared, and the Authors made an effort to thoroughly explain the elements of the research process. Comments:

  1. In Table 1, the range of results should be provided for IQ classification.
  2. Please indicate the age group for which the LEC scale has been developed.

Results

The method of developing and presenting the results does not raise any doubts.

Discussion

The Authors show the importance of their research against the background of relevant research results to date. In Future research, it is worth signaling the need to analyze the importance of the type of disorder for the occurrence of ACEs, especially some of their categories.

Reviewer 2 Report

Article of great interest to the academic community. It is advisable to carry out a general review of the writing in English.

Reviewer 3 Report

Paper is well constructed and methodologically sound. Intro and bibliography are well done.

Recommendations:

  1. Mention what program was used for statistical analysis at the beginning of the methodology section.
  2. Demographics: Knowing what I know about EBD, I would think some stats might be skewed right. Prove me wrong by also providing either the mean or skew/kurtosis for IQ.
  3. Do tests have an omnibus score? Subscales? Relationship by matching between scales?
  4. In the appendix, for each table, list what test was used (otherwise one must find the information in the text).

Major problem: Lines 405-408 compare your findings with previous research, concluding EBD students in your study have a much higher prevalence rate. The results are from a faulty comparison and might not be true. Why? Lines 406-408 rely on checking for 10 ACEs compared to you examining 25 ACEs (according to lines 256-257). Unless you directly compare only the 10 in each (which could include, if described, a concatenation of the 25 to 10), checking for 25 variables might be the noise confounding your results. A natural conclusion is an expansion of a definition, a broadening of the spectrum, might mean your inclusivity produced the results. 

Round 2

Reviewer 3 Report

I think your article is valuable and much clearer. You also expand on the initial view of ACEs. Most authors would've resisted such improvements; yet, if you hadn't, and I had read your article after publication for the first time, I would have post-reviewed your article for a possible logical fallacy, calling into question your entire paper. Kudos for going above and beyondl

A pedantic question: You list ASS as an abbreviation. Is that really used in English?

Author Response

Kind regards,

Evelyne Offerman 
